# Proteomic Changes in *Paspalum fasciculatum* Leaves Exposed to Cd Stress

**DOI:** 10.3390/plants11192455

**Published:** 2022-09-20

**Authors:** Manuel Salas-Moreno, María Ángeles Castillejo, Erika Rodríguez-Cavallo, José Marrugo-Negrete, Darío Méndez-Cuadro, Jesús Jorrín-Novo

**Affiliations:** 1Biosistematic Research Group, Biology Department, Faculty of Basic Sciences, Technological University of Chocó, Quibdó 270002, Colombia; 2Analytical Chemistry and Biomedicine Group, Department of Biology, University of Cartagena, Cartagena 130015, Colombia; 3Agroforestry and Plant Biochemistry, Proteomics, and Systems Biology Research Group, Department of Biochemistry and Molecular Biology—ETSIAM, University of Cordoba, UCO-CeiA3, 14071 Cordoba, Spain; 4Water, Applied and Environmental Chemistry Group, Chemistry Department, Faculty of Basic Sciences, University of Córdoba, Montería 230002, Colombia

**Keywords:** *Paspalum fasciculatum*, Cd stress, changes in the proteome, downregulated, upregulated

## Abstract

(1) Background: Cadmium is a toxic heavy metal that is widely distributed in water, soil, and air. It is present in agrochemicals, wastewater, battery waste, and volcanic eruptions. Thus, it can be absorbed by plants and enter the trophic chain. *P. fasciculatum* is a plant with phytoremediation capacity that can tolerate Cd stress, but changes in its proteome related to this tolerance have not yet been identified. (2) Methods: We conducted a quantitative analysis of the proteins present in *P. fasciculatum* leaves cultivated under greenhouse conditions in mining soils doped with 0 mg kg^−1^ (control), 30 mg kg^−1^, or 50 mg kg^−1^. This was carried out using the label-free shotgun proteomics technique. In this way, we determined the changes in the proteomes of the leaves of these plants, which allowed us to propose some tolerance mechanisms involved in the response to Cd stress. (3) Results: In total, 329 variable proteins were identified between treatments, which were classified into those associated with carbohydrate and energy metabolism; photosynthesis; structure, transport, and metabolism of proteins; antioxidant stress and defense; RNA and DNA processing; and signal transduction. (4) Conclusions: Based on changes in the differences in the leaf protein profiles between treatments, we hypothesize that some proteins associated with signal transduction (Ras-related protein RABA1e), HSPs (heat shock cognate 70 kDa protein 2), growth (actin-7), and cellular development (actin-1) are part of the tolerance response to Cd stress.

## 1. Introduction

Cadmium (Cd) is a non-essential element that is toxic to living organisms. It is fatal at doses of 1500 to 8900 mg. It is widely distributed in various environmental matrices, and is emitted by internal combustion vehicles and friction elements such as brakes and tires [1,2,3]. Cd is also present in the soil due to the excessive use of agrochemicals in agriculture, wastewater, volcanic eruptions, battery waste, and acid rain. In this way, it can be absorbed by plants and, consequently, enter the trophic chain. Cd uptake can cause physiological changes in plant growth, biomass accumulation, photosynthesis, water absorption, nutrients, and cell transport, and can lead to redox control activity disorders and cell death [4,5,6,7].

However, some plants can grow under this abiotic stress, as they are able to absorb and tolerate the presence of heavy metals such as Cd in their tissues at levels that would be toxic to other plants. These plants are called phytoremediators, and they are used to remove metals from contaminated soils.

To resist toxic effects, phytoremediation plants must make adjustments to their proteomes that involve changes at the molecular signaling and gene expression regulation levels. In general, changes in the plant proteome begin with the molecular perception of stress, followed by the activation of signal transduction and the expression of a great variety of genes induced by stress [8,9,10]. At the protein level, the overproduction of heat shock proteins (HSPs) occurs during abiotic stress, as they act as chaperones, protecting proteins that have lost their native conformation or have been misfolded during a stress event, and helping to eliminate and degrade the non-corrected proteins [11,12,13]. 

Another effect of Cd stress occurs through the production of reactive oxygen species (ROS), which cause damage to proteins and other biomolecules, such as lipids and DNA. In response to an increase in ROS, the overproduction of catalase, ascorbate peroxidase, superoxide dismutase, peroxidase, or compounds such as ascorbate and glutathione is frequently observed at the tissular level [14,15,16,17].

The changes in protein abundance under metal stress conditions should be determined to characterize specific adaptive responses in phytoremediator plants. Different quantitative and qualitative proteomics techniques have been used alone or in combination to elucidate the effects of Cd stress in different plant species, e.g., *Microsorum pteropus* [15], *Brachypodium distachyon* [3], *Arabidopsis thaliana* [18], *Oryza sativa* [19,20], *Populus yunnanensis* [21], *Brassica juncea* [22], *Brassica napus* [23], and *Triticum aestivum* L. [24].

In particular, we focus on *Paspalum fasciculatum* Willd. Ex. Flüggé—a plant of the Poaceae family that is widely distributed in tropical and subtropical ecosystems, and can grow in mining soils with high levels of contamination by the heavy metals Cd and Pb thanks to its phytoremediation and phytoextraction characteristics [25]. Similar to other plants, it also exhibits an antioxidant response, limiting the oxidative damage to its proteome promoted by these metals [26].

In order to contribute to the elucidation of the mechanisms of Cd toxicity tolerance, we conducted a shotgun (LC–MS/MS) proteomics analysis to study changes in the leaf proteome of *P. fasciculatum* Willd. Ex. Fluggé plants grown under greenhouse conditions in mining soils doped with this heavy metal.

## 2. Results

The physical and chemical characteristics of the soil samples, the concentration of Cd in the tissues, and the growth behavior of *P. fasciculatum* were established in a previous work [26]. The concentrations in the leaves of the plants ranged from 4.2 to 55.72 mg kg^−1^, and were measured over periods of 30, 60, and 90 days. The roots showed three times higher concentrations of Cd than the other organs (66.08 to 367.98 mg kg^−1^), clearly showing that they represent the mechanism by which exclusion of heavy metals is carried out.

Additionally, a higher growth rate was observed in the plants that underwent treatments TC30 (30 mg kg^−1^ Cd) and TC50 (50 mg kg^−1^ Cd) with respect to the control group (*p* ≤ 0.05) in the first 60 days. Due to this behavior, the 60-day-old plants were selected for the proteomics analysis, and the results obtained are shown.

### 2.1. Quantitative Analysis of Proteins in P. fasciculatum Leaves Exposed to Cd Stress

The proteins extracted from *P. fasciculatum* leaves subjected to treatments TC30, TC50, and TC (control) were subjected to a quantitative proteomics analysis using label-free liquid chromatography–tandem mass spectrometry (LC–MS/MS) shotgun proteomics (Table 1). The raw mass spectral data were analyzed using the SEQUEST search engine. A total of 1227 proteins was identified, of which 329 met the parameters and inclusion criteria for the quantitative analysis (i.e., high confidence for the identification of peptides, XCorr ≥ 2, and at least two different peptides per protein). In TC30, 175 proteins were differentially abundant compared with the control—77 were upregulated, while 98 were downregulated—whereas in TC50, 74 were upregulated, and 143 were downregulated (Appendix A). However, only 90 of these proteins showed statistically significant differences (*p* ≤ 0.05), and 46 proteins showed the same behavior in both treatments (Table 2).

The proteins involved in the processes of photosynthesis and energy metabolism were the most commonly downregulated by Cd stress. On the other hand, proteins with molecular functions such as heat shock, antioxidant activity, stress response, transduction signals, and cell development and growth were the most frequently upregulated (Table 2). This can explain the tolerance of these plants and the significant biomass accumulated in comparison to the controls.

### 2.2. Functional Classification of Proteins

Proteins with significant changes in accumulation (Figure 1) were classified into eight main groups: (i) proteins involved in carbohydrate and energy metabolism (80); (ii) proteins involved in photosynthesis (71); (iii) proteins involved in protein structure, transport, and metabolism; (iv) proteins involved in antioxidant defense and stress (56); (v) proteins involved in growth, development, and cellular organization (13); (vi) proteins involved in DNA and RNA processing (18); (vii) signal transduction proteins (14); and (viii) proteins with no known functions (5). The functions of the proteins involved in the tolerance mechanisms of *P. fasciculatum* are explained in greater detail in the Discussion.

## 3. Discussion

The changes in the proteomes of plants with phytoremediation characteristics or tolerance to Cd stress have been the subject of many investigations, because knowledge in this area could contribute to the elucidation of the molecular resistance mechanisms selected in them [17,20,22,24]. *P. fasciculatum* is a plant with a Cd phytostabilization capacity. It could be a promising alternative for the removal of Cd from contaminated soils or areas degraded by mining. Therefore, we studied the changes in the proteome of *P. fasciculatum* leaves using label-free LC–MS/MS shotgun proteomics experiments, with the aim of delving into the metabolic pathways altered by the presence of Cd and describing, for the first time, the tolerance mechanisms that this phytoremediation species has developed in response to Cd stress.

To explain these tolerance mechanisms, we focused on the behaviors of the 90 proteins that showed statistically significant changes in abundance in *P. fasciculatum* leaves in the TC30 (30 mg kg^−1^ Cd) and TC50 (50 mg kg^−1^ Cd) treatment groups compared with the TC group (Table 2). These mechanisms included an increase in the abundance of proteins involved in the photosynthesis, energy metabolism, signal transduction, antioxidant defense, protein folding and degradation, and cellular growth and development pathways (Figure 2).

### 3.1. P. fasciculatum Overcomes the Deleterious Effects of Cadmium on Photosynthesis Proteins and Carbohydrate Energy Metabolism

At day 60, the proteins involved in photosynthesis were affected by Cd stress. Fifteen of these decreased in both exposed groups (TC30 and TC50), while another fifteen belonging to the carbohydrate and energy metabolism protein groups decreased in the TC50 group.

Photosynthesis is a vital process for plant growth and development that plays important roles in carbon fixation and energy metabolism [17]. In our study, photosynthesis proteins were highly affected. Seventeen proteins were downregulated and six were upregulated. One of the most abundant proteins in nature, which is vital for the photosynthesis process, is the enzyme ribulose-1,5-bisphosphate carboxylase/oxygenase large subunit (RuBisCO LSU), as well as ribulose bisphosphate carboxylase/oxygenase activase A. These enzymes decreased by 7.2-fold (TC30 and TC50) and 1.5-fold (TC50), respectively.

RuBisCO participates in carbon dioxide (CO_2_) assimilation reactions in plants. It is a transketolase enzyme that catalyzes the first step of the Calvin cycle and the reaction that produces the CO_2_ acceptor molecule ribulose-1,5-bisphosphate (Figure 2) [27]. Similar studies carried out on plants, such as *Phytolacca americana* [7] and *Spinacia oleracea* L. [28], demonstrated that Cd inhibits the activity of RuBisCO. However, in young *Theobroma cacao* plants, Cd increased the abundance of RuBisCO by sevenfold compared to controls [11]. Therefore, these results show that Cd stress has significant adverse effects on the efficiency of carbon fixation, which could be related to chlorophyll damage [17]. In agreement with the above, four chlorophyll-binding proteins were downregulated in both treatments: the chlorophyll-binding protein ab 25 (8.3-fold), the chlorophyll-binding protein ab (fragment) (9.1-fold), the protein of chlorophyll ab type 2 member 1B binding (12.1-fold), and the chlorophyll ab binding protein of LHCII type I (12.5) (Figure 2). These are the Apo proteins of the light-capturing complex that capture and deliver excitation energy to photosystems I and II (PSI and PSII) [29].

Our results are similar to those produced by a study on *Arabidopsis halleri*, which demonstrated inhibited and decreased activity due to Cd stress in PS II and I [30]; however, in Cd-tolerant species such as *P. yunnanensis* [21] and *Zygophyllum fabago* L. [31], an increase in chlorophyll a–b binding protein has been observed. In PSI and PSII, chlorophyll a–b binding proteins are essential in the oxygenated photosynthesis processes; therefore, the downregulation of these proteins could significantly compromise oxygen production [32]. In addition, Cd can replace metal ions such as Fe^2+^, Ca^2+^, and Mn^2+^, which can affect the reaction center of PSII, altering the oxidant system of PSII water. In this way, electron transport can be uncoupled in chlorophyll, generating significant amounts of ROS, such as superoxide anions (O_2_^•−^) and hydrogen peroxide (H_2_O_2_) [33,34,35]. The adverse effects described for these photosynthesis proteins due to Cd stress are in accordance with the oxidative damage reported in RuBisCO LSU proteins and the chlorophyll a–b binding protein CP26 in the leaves of *P. fasciculatum* [26].

Despite the inhibitory effects on photosynthesis proteins, *P. fasciculatum* seedlings grew adequately, without showing any apparent toxic effects, such as necrosis or chlorosis. This indicates that the plant activates mechanisms that allow it to repair its photosynthetic structures and guarantee the energetic supply during exposure to Cd. In this sense, eight upregulated proteins were found in the *P. fasciculatum* leaves exposed to Cd. One of these—ATP synthase subunit beta (3.5-fold in TC50)—plays a role in the production of energy across the thylakoid membrane through the production of ATP using a proton gradient [36]. Glucose-1-phosphate adenylyltransferase small subunit (8.1-fold TC50), phosphoglycerate kinase 1 (12-fold TC50), and RuBisCO large subunit-binding protein subunit beta (2.7- and 7.3-fold in TC30 and TC50, respectively) are important enzymes involved in the Calvin cycle and carbohydrate biosynthesis. Other upregulated proteins were chlorophyll a–b binding protein P4 (10.7- and 2.5-fold in TC30 and TC50, respectively), photosystem II protein D1(PII-D1) (14- and 13.9-fold in TC30 and TC50, respectively), and ATP-dependent zinc metalloprotease FtsH (FstH) (7.8- and 8.2-fold in TC30 and TC50, respectively) (Figure 2). FstH plays an essential role in the formation of thylakoid membranes during the early development of chloroplasts, as well as controlling the quality of proteins during the photosynthesis process [37]. Therefore, the activity of this protein may be important for the development and maintenance of the functioning of chloroplasts, as well as in the repair or renewal of photosynthesis proteins affected by Cd stress. FtsH also participates in the PSII repair cycle, causing the degradation of PII-D1 in photosystem II and its subsequent de novo synthesis [37], as shown by the upregulation of PII-D1.

The proteins involved in glycolysis, gluconeogenesis, and starch biosynthesis were strongly downregulated, in a range of 1.5–10.7-fold lower than the control (Table 2). Such proteins included phosphoglucomutase, triosephosphate isomerase, glyceraldehyde-3-phosphate dehydrogenase, glyceraldehyde-3-phosphate dehydrogenase 1, glyceraldehyde-3-phosphate dehydrogenase 2, enolase 2, fructokinase-1, and phosphoenolpyruvate carboxykinase (ATP) 2 (Figure 2). These proteins are essential for the synthesis of metabolic intermediates and amino acids and fructose metabolism, as well as for the production of energy [17,38,39]. However, in species such as *Microsorum pteropus*, *T. cacao,* and *Brachypodium distachyon*, under conditions of Cd stress, the upregulation of proteins involved in energy metabolism has been reported [3,11,17]. On the other hand, in similar studies carried out on *Cannabis sativa*, the downregulation of glycolysis and gluconeogenesis proteins was observed [40]. In our case, only two proteins were upregulated in TC30: the putative aconitate hydratase (ACO), and enolase (ENO 1).

ACO is an important enzyme in the Krebs cycle and the glyoxylate cycle; in leaves, it is essential for citrate flux in the cytosol [41]. This enzyme plays an important role in the accumulation of resistance to oxidative stress and cell death in *A. thaliana* and *Nicotiana benthamiana* [42]. ENO1, a key enzyme in glycolysis, catalyzes the conversion of 2-phosphoglycerate into phosphoenolpyruvate. It plays an important role in energy metabolism in cells. In other studies on tolerance to Cd stress in *B. distachyon*, upregulation of ENO1 was observed [3].

In terms of proteins involved in the Krebs cycle, four downregulated proteins were observed. This cycle comprises several chemical reactions that can store metabolic energy through the oxidation of acetyl-CoA from carbohydrates, fats, and proteins into CO_2_ and the formation of adenosine triphosphate (ATP) [17].

ATP synthase subunit alpha, cytochrome c, malate dehydrogenase 1, and NADH dehydrogenase (ubiquinone) iron–sulfur protein 8 were shown to be downregulated in this study. The downregulation of Krebs cycle proteins has also been observed in experimental studies on *C. sativa* [40]. However, some species that are tolerant to Cd stress, such as *M. pteropus* and *B. distachyon*, presented significant upregulation of the proteins involved in this cycle. Thus, a large part of their resistance to Cd stress was predicted to be based on these proteins [3,17]. Only pyruvate dehydrogenase E1 component subunit beta-1 was shown to be upregulated by 8.9-fold in TC30. This protein could be a key element in ATP homeostasis in *P. fasciculatum*, accelerating the production of pyruvate and acetyl-CoA, and reducing NAD^+^ to NADH to increase ATP synthesis [43].

The proteins related to photosynthesis, energy metabolism, and protein metabolism showed strong downregulation; however, *P. fasciculatum* showed significant growth (*p* < 0.05) for 60 days, without showing adverse effects such as chlorosis or necrosis (Appendix A). The upregulation of some proteins (already described above) in these metabolic pathways is key to improving photosynthetic activity and energy production. However, with the changes in the proteome identified in the *P. fasciculatum* leaves, it can be clearly seen that multiple mechanisms are used to enhance the activity of various metabolic pathways in response to Cd stress, which we describe below.

### 3.2. Signal-Transduction-Associated Protein Changes during Cd Stress

Proteins involved in signal transduction can experience increases in their abundance levels when plants are under conditions of abiotic stress, e.g., due to exposure to Cd. The 14-3-3 proteins regulate the activities of some proteins involved in signal transduction [3]; these proteins may play important roles in the response to and defense against abiotic and biotic stressors in plants [44]. The 14-3-3 proteins participate in specific protein phosphorylation cascades [45]. In this way, their enzymatic activity can be altered. They can also prevent or induce protein degradation. Likewise, this can affect the locations of proteins within cell organelles by binding to their targets [46,47]. Furthermore, these proteins can activate Ca^2+^-dependent protein kinase (CDPK), while ROS-like H_2_O_2_ can also activate CDPK by increasing cytosolic Ca^2+^. In this way, they can activate transcription factors that regulate the expression levels of specific genes (Figure 2) [48]. These proteins play important roles in the tolerance mechanisms of plants. They allow the cells to modify metabolic pathways, such as those associated with defense, photosynthesis, and redox homeostasis [49]. In this study, seven differentially upregulated proteins involved in signal transduction were identified in *P. fasciculatum* leaves in both treatments, three of which were from the 14-3-3 protein family: the 14-3-3 -like protein (2.1-fold in TC50), the 14-3-3-like protein GF14-B (2.2-fold in TC50), and the 14-3-3-like protein GF14-D (2.7- and 2.4-fold in TC30 and TC50, respectively). Of the multiple mechanisms of tolerance to Cd stress associated with *P. fasciculatum*, the 14-3-3 proteins play a fundamental role, because these proteins interact with other proteins involved in cell signaling, energy metabolism, antioxidant defense, and protein folding and repair.

Other important proteins involved in cell signaling identified in this study included the Rab proteins, which also play important roles in vesicular traffic. However, these proteins are also important for the detoxification, transport, and compartmentalization processes of heavy metals in vacuoles [50]. In this study, Ras-related protein Rab-2-B (Rab-2; 6.4- and 7.3-fold in TC30 and TC50, respectively), Ras-related protein RABE1c (Rabe1c; 1.6-fold in TC50), Ras-related protein RABA1e (Raba1e; 8.6- and 7.6-fold in TC30 and TC50, respectively), and Ras-related protein Rab7 (Rab7; 6.3-fold in TC50) were upregulated. Experimental studies have shown that Rab7 can be transported to the vacuoles by vesicular transport [51]. This suggests that these proteins could be involved in the transport of Cd to vacuoles (Figure 2). Similar results have been reported for *A. thaliana* plants exposed to aluminum stress [52]. In addition, it is known that these proteins play important roles in the growth and development of plants, and play a key role in the response to biotic and abiotic stress [51].

### 3.3. Protein Changes Associated with Growth and Development during Cd Stress

Actins are essential proteins that have functions in the composition of the cytoskeleton, cell division and growth, morphogenesis, and hormonal transport [53,54,55]. In this study, five types of these proteins were upregulated: actin (3.0-fold in TC50), actin-1 (8.4- and 9.2-fold in TC30 and TC50, respectively), actin-depolymerizing factor 2 (3.0-fold in TC30), actin-depolymerizing factor 3 (3.0-fold in TC50), and actin-7 (1.5- and 7.6-fold in TC30 and TC50, respectively). These proteins were shown to be important for the growth and development of seedlings exposed to Cd stress for 60 days, and were likely responsible for the significant growth of these plants compared to the controls.

### 3.4. Antioxidant-Defense- and Stress-Response-Involved Protein Changes during Cd Stress

Other important proteins involved in the multiple mechanisms of tolerance of *P. fasciculatum* that were upregulated included the heat shock proteins (HSPs): heat shock 70 kDa protein, heat shock cognate 70 kDa protein 2, 17.0 kDa class II heat shock protein, and 18.2 kDa class I heat shock protein. These proteins accumulate in organisms such as plants to allow them to adapt genetically and eco-physiologically to stressful environments [56]. HSPs are likely to play a role in the repair of proteins damaged by Cd stress in *P. fasciculatum* leaves (Figure 2). These proteins have the ability to protect and repair proteins under abiotic stress conditions. They aid in folding and degradation, and act on misfolded proteins that have lost their normal assembly [3]. ROS are responsible for a large part of the damaging effects on proteins caused by Cd stress in *P. fasciculatum* leaves [25].

The uptake of Cd into cells by zinc-regulated, iron-regulated transporter-like protein (ZIP) [57] can generate ROS via the Haber–Weiss/Fenton reaction through the displacement of cations, or by activating NAPH oxidase (Figure 2) [58]. Under Cd stress conditions, these plants have an antioxidant defense system. Antioxidant enzymes are involved in this system, and are crucial to the elimination of ROS [59]. The proteins involved in the antioxidant defense were also identified in *P. fasciculatum* leaves; however, Cd stress caused the downregulation of four proteins. Only peroxiredoxin 1 was upregulated, by 2.2- and 2.4-fold in TC30 and TC50, respectively (Table 2). These proteins play roles in the response to stress, auxin catabolism, lignification, and H_2_O_2_ removal, and are secreted by vacuoles, apoplasts, or cell walls [60]. Peroxidase 1 plays an important role in the detoxification of toxic H_2_O_2_ accumulated in the vacuoles due to Cd stress in *P. fasciculatum* (Figure 2). For plant cells to grow and develop properly, it is crucial to eliminate these toxic molecules [61].

In summary, based on different changes in the leaf protein profiles among the treatment groups, we hypothesize that some proteins involved in signal transduction (Ras-related protein RABA1e), growth (actin-7), and cellular development (actin-1), as well as HSPs (heat shock cognate 70 kDa protein 2), are part of the tolerance response to Cd stress, as illustrated in Figure 2.

## 4. Materials and Methods

### 4.1. Plant Growth Conditions and Tissue Sampling

*P. fasciculatum* plants were propagated from cuttings (5 cm) obtained from individuals of 1 m in length and grown in metal-free soil [25,26]. Three weeks after cut-sowing, plantlets were transferred to the soil from the “Alacrán” mine (northwestern Colombia, 7°44′29.0″ N and 75°44′10.8″ W) (Appendix A). They were supplemented or not (control plants) with two concentrations of Cd: 30 and 50 mg kg^−1^. The preparation and the physical and chemical characteristics of the soil samples were reported by Salas Moreno et al. [26]. The experiment was performed under greenhouse conditions with an average temperature of 25–30 °C, a relative humidity of between 55 and 65%, and a natural illumination of around 12 h. Throughout the 60-day experiment, the plants were visually monitored, and the biomass and metal contents were determined in the roots, stems, and leaves [26]. At day 60, leaves from the control and Cd-treated plants were picked, abundantly washed with water, dried with filter paper, frozen in liquid nitrogen, and lyophilized. The experimental design consisted of three replicates per treatment (control, 30 mg kg^−1^, and 50 mg kg^−1^ soil of Cd). Each replicate contained leaves from *P. fasciculatum* plants, corresponding to 1 g of fresh weight. In Appendix A, a photograph of the plants used in each treatment group is included.

### 4.2. Protein Extraction and Digestion

Lyophilized leaf tissue (1000 mg weight) was ground to a fine powder in liquid nitrogen using a pestle and mortar. Proteins were extracted from the powder by using the TCA–acetone–phenol protocol, as reported by Wang et al. [62], and then quantified with the Bradford assay, using bovine serum albumin (BSA) as the standard [63].

Protein samples (50 µg equivalent of BSA) were cleaned and concentrated by SDS–PAGE (12%), with the gel being Coomassie-stained [64]. The single band obtained was manually excised, destained, and digested with proteomics-grade trypsin (Promega, Madison, WI, USA) to a final concentration of 12.5 ng µL^−1^, in accordance with the method used by Romero-Rodríguez et al. [65].

Digested peptides were desalted using C18 cartridges (Scharlau, Barcelona, Spain), and eluted peptides were vacuum-dried and dissolved in a mixture of 70:30 (*v*/*v*) acetonitrile (ACN)/water containing 0.1% trifluoroacetic acid.

### 4.3. Shotgun LC–MS/MS Analysis

Nano-LC was performed using a Dionex Ultimate 3000 nano UPLC (Thermo Scientific, Waltham, MA, USA) with a C18 75 µm × 50 Acclaim PepMap column (Thermo Scientific). The peptide mix was previously loaded onto a 300 µm × 5 mm Acclaim PepMap precolumn (Thermo Scientific) in 2% AcN/0.05% TFA for 5 min at 5 µL/min. Peptide separation was performed at 40 °C for all runs. Mobile phase A was 0.1% formic acid, and mobile phase B was 80% AcN, 0.1% formic acid. Samples were separated over 60 min using a gradient ranging from 96% A to 90% B and a flow rate of 300 nL/min. Eluted peptides were converted into gas-phase ions by nano-electrospray ionization and analyzed using a Thermo Orbitrap Fusion (Q-OT-qIT, Thermo Scientific) mass spectrometer operated in positive mode. Survey scans of peptide precursors were performed from 400 to 1500 *m*/*z* at a 120 K resolution (at 200 *m*/*z*), with a 4 × 10^5^ ion count target. Tandem MS was performed by isolation at 1.2 Da with the quadrupole, CID fragmentation with a normalized collision energy of 35, and rapid scan MS analysis in the ion trap. The AGC ion count target was set to 2 × 10^3^, and the maximum injection time was 300 ms. Only precursors with a charge state of 2–5 were sampled for MS^2^. The dynamic exclusion duration was set to 15 s, with a 10 ppm tolerance around the selected precursor and its isotopes. Monoisotopic precursor selection was turned on. The instrument was run in top 30 mode with 3 s cycles, meaning that the instrument would continuously perform MS^2^ events until achieving a maximum of 30 top non-excluded precursors or a duration of 3 s—whichever was shorter.

### 4.4. Protein Identification and Functional Classification

The raw data derived from the MS experiments were processed using Proteome Discoverer software v.1.4 (Thermo Scientific). MS^2^ spectra were searched using the SEQUEST engine against the protein FASTA files created from a combination of the following databases: UniProtKB and the Swiss-Prot database, with taxonomy restrictions to Viridiplantae. The precursor mass tolerance was set to 10 ppm, and the fragment ion mass tolerance was fixed to 0.8 Da. Only charge states of +2 or greater were used. The identification confidence was set to 5% FDR, and acetylation of the N-terminus, oxidation of methionine, and carbamidomethyl cysteine formation were set as variable modifications. No fixed modifications were set. Trypsin was set as a proteolytic enzyme, and a maximum of two miscleavages was set for all searches. A minimum XCorr of 2 and proteins with 3 or more peptides matched were considered. For relative quantification, the peak area of identified peptides was used.

The protein peak areas were normalized by the total sum of the peak area values per sample, and missing values were corrected. Mean values and standard deviations (SDs) of the peak areas of protein species were determined using three independent analyses (Appendix A). The proteins determined were identified, and their functional classifications were established with the help of the UniProtKB database.

To determine significant quantitative differences in protein abundance, normalized ratio values between treatments (TC30/C, TC50/C) ≥ 1.5 (upregulated) or ≤0.67 (downregulated) and *p*-values of 0.05 were considered. Biological significance (log2(fold change)) was used to describe the changes induced by Pb treatments on the *P. fasciculatum* leaf proteome (proteins with more/less than 1.5-fold changes in abundance levels; Table 2). Differentially accumulated proteins were classified based on Gene Ontology terms annotated in the UniProtKB protein database using molecular functions and biological processes.

### 4.5. Statistical Analysis

For the experiments (protein peak areas) with *P. fasciculatum* in mining soil, the results are presented as the mean ± the standard deviation of triplicate determinations. After normalization of the protein peak areas, the data were subjected to ANOVA and, when necessary, a comparison of means was performed using the Bonferroni test. The statistical software GraphPad Prism version 8.0.1 was used for all analyses. Results were considered to be significant at the *p* ≤ 0.05 level.

## 5. Conclusions

*P. fasciculatum* is a Cd phytoremediation plant that has demonstrated a Cd phytostabilization capacity. The results of this research demonstrate important changes in the proteome of *P. fasciculatum* leaves in their adaptation to Cd-doped mining soils. These changes were mainly observed in proteins involved in energy, protein metabolism, photosynthesis, signal transduction, and cell growth and development, as well as HSPs. Adverse changes in protein abundance were observed in photosynthesis and carbohydrate metabolism; however, these changes were not shown to be a threat to plant growth and development, which could be supported by the upregulation of proteins such as actin, actin-1, actin-depolymerizing factor 2, actin-depolymerizing factor 3, and actin-7. The results of this study may prompt new research aimed at the response to heavy metal stress in phytoremediation plants in areas degraded by gold mining.

## Figures and Tables

**Figure 1 plants-11-02455-f001:**
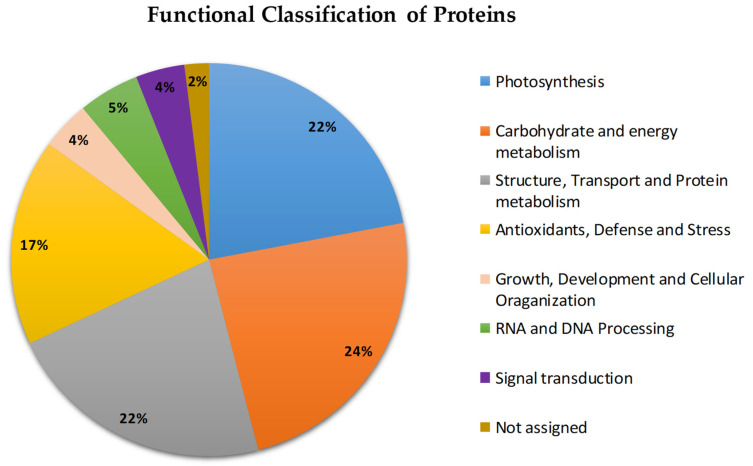
Functional classification of the proteins identified in *P. fasciculatum* leaves exposed to Cd stress for 60 days.

**Figure 2 plants-11-02455-f002:**
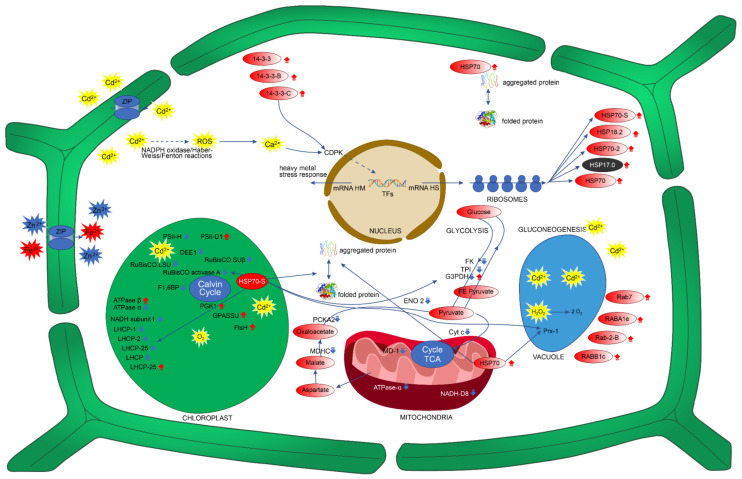
Possible mechanisms of tolerance in response to Cd stress based on changes in the proteome in *P. fasciculatum* leaves after 60 days. RuBisCO LSU: ribulose bisphosphate carboxylase large chain; RuBisCO SUβ: RuBisCO large subunit-binding protein subunit beta; RuBisCO activase A: ribulose bisphosphate carboxylase/oxygenase activase A; LHCP-25: chlorophyll a–b binding protein 25; LHCP-2: chlorophyll a–b binding protein type 2 member 1B; LHCP-1: chlorophyll a–b binding protein of LHCII type I; LHCP4: chlorophyll a–b binding protein P4; MDHC: malate dehydrogenase; PCKA2: phosphoenolpyruvate carboxykinase (ATP) 2; PSII-H: photosystem II reaction center protein H; PSII D1: photosystem II protein D1; NADH subunit I: NAD(P)H-quinone oxidoreductase subunit I; GPASSU: glucose-1-phosphate adenylyltransferase small subunit; Cyt c: cytochrome c; ATPase α: ATP synthase subunit alpha; NADH-D8: NADH dehydrogenase (ubiquinone) iron–sulfur protein 8; PE Pyruvate: phosphoenolpyruvate; TPI: triosephosphate isomerase; FK1: fructokinase-1; GEPDH: glyceraldehyde-3-phosphate dehydrogenase 1; PGK1: phosphoglycerate kinase 1; ENO2: enolase 2; MD-1: malate dehydrogenase 1; Prx-1: peroxidase 1. HSP17.0: 17.0 kDa class II heat shock protein. HSP18.2: 18.2 kDa class I heat shock protein; HSP70: heat shock cognate 70 kDa protein; HSP70-2: heat shock cognate 70 kDa protein 2; HSP70S: stromal 70 kDa heat-shock-related protein; FtsH: ATP-dependent zinc metalloprotease, FtsH; Rab7: Ras-related protein Rab7; RABA1e: Ras-related protein RABA1e; Rab-2-B: Ras-related protein Rab-2-B; RABB1c: Ras-related protein RABB1c; 14-3-3: 14-3-3-like protein; 14-3-3-B: 14-3-3-like protein GF14-B; 14-3-3-D: 14-3-3-like protein GF14-D; CDPK: Ca^2+-^ dependent protein kinase; TFs: transcription factors; PE Pyruvate: phosphoenolpyruvate; ZIP: zinc-regulated, iron-regulated transporter-like protein (ZIP). Red arrow = upregulation; blue arrow = downregulation; mRNA HS: mRNA of heat shock protein; mRNA MH: mRNA in response to heavy metals.

**Table 1 plants-11-02455-t001:** Protein yield from *Paspalum fasciculatum* leaves exposed to Cd stress for 60 days (*n* = 3).

Treatments	Fresh Weight of Protein (mg g^−1^) ± SD
TC ^1^	2.23 ± 0.2
TC30 ^2^	2.43 ± 0.60
TC50 ^3^	2.54 ± 0.35

^1^ TC: control, ^2^ TC30: 30 mg kg^−1^, ^3^ TC50: 50 mg kg^−1^.

**Table 2 plants-11-02455-t002:** Proteins identified in *Paspalum fasciculatum* leaves after 60 days of exposure to 30 and 50 mg kg^−1^ Cd.

No.	Accession ^1^	Protein Name	MW (kDa) ^2^	Coverage ^3^	Source	Fold Change ^4^ TC30/TC TC50/TC		Subcellular ^5^ Localization
*Photosynthesis*								
1	B1NWD5	ATP synthase subunit alpha	55.564	29,7830374	*Manihot esculenta*	0.4	−7.9	Chloroplast
2	P04782	Chlorophyll a–b binding protein 25	28.149	28.1954887	*Petunia* sp.	−8.3	−8.3	Chloroplast
3	P12332	Chlorophyll a–b binding protein (fragment)	22.029	19.0243902	*Silene latifolia* subsp. *alba*	−9.1	−9.1	Chloroplast
4	P15194	Chlorophyll a–b binding protein type 2 member 1B	29.014	31.3868613	*Pinus sylvestris*	−12.2	−12.2	Chloroplast
5	P12328	Chlorophyll a–b binding protein of LHCII type I	28.359	15.1515152	*Lemna gibba*	−12.5	−12.5	Chloroplast
6	A1E9K3	Cytochrome f	35.341	49.0625	*Hordeum vulgare*	−5.9	−5.9	Chloroplast
7	P69390	Cytochrome b559 subunit alpha	9.439	39.7590361	*Hordeum vulgare*	−3.9	−2.6	Chloroplast
8	O64422	Fructose-1,6-bisphosphatase	43.577	26.8472906	*Oryza sativa* subsp. *japonica*	−0.3	−1.9	Chloroplast
9	Q01516	Fructose-bisphosphate aldolase 1 (fragment)	38.633	15.7303370	*Pisum sativum*	−4.2	−4.2	Chloroplast
10	P17606	Malate dehydrogenase (NADP) 1	46.426	17.4825174	*Sorghum bicolor*	0.1	−8.1	Chloroplast
11	B3TNA5	NAD(P)H-quinone oxidoreductase subunit H	45.795	30.2798982	*Brachypodium distachyon*	−7.4	−7.4	Chloroplast
12	P46722	NAD(P)H-quinone oxidoreductase subunit I	21.143	51.1111111	*Zea mays*	−10.0	−10.0	Chloroplast
13	O49079	Oxygen-evolving enhancer protein 1	34.848	21.8844984	*Fritillaria agrestis*	−0.3	−1.6	Chloroplast
14	Q09ME8	Photosystem II reaction center protein H	7.664	23.2876712	*Citrus sinensis*	−9.4	−9.4	Chloroplast
15	Q40073	Ribulose bisphosphate carboxylase/oxygenase activase A	51.041	20.4741379	*Hordeum vulgare*	−0.5	−1.5	Chloroplast
16	Q37227	Ribulose bisphosphate carboxylase large chain (fragment)	49.103	28.8939051	*Iris germanica*	−7.2	−7.2	Chloroplast
17	O23813	Sulfite reductase (ferredoxin)	69.971	19.0551181	*Zea mays*	−4.3	−12.5	Chloroplast
18	Q0P3P2	ATP synthase subunit beta	51.83	21.9461697	*Ostreococcus tauri*	0.3	3.5	Chloroplast
19	P55240	Glucose-1-phosphate adenylyltransferase small subunit (fragment)	13.238	20	*Zea mays*	0	8.1	Chloroplast, Amyloplast
20	Q9SQL2	Chlorophyll a–b binding protein P4	27.212	18.2539683	*Pisum sativum*	10.7	2.5	Chloroplast
21	Q9LD57	Phosphoglycerate kinase 1	50.081	25.3638253	*Arabidopsis thaliana*	0	12.0	Chloroplast
22	A6YGB8	Photosystem II protein D1	38.125	18.3139535	*Pleurastrum terricola*	14.0	13.9	Chloroplast
23	P08927	RuBisCO large subunit-binding protein subunit beta	62.945	21.3445378	*Pisum sativum*	2.7	7.3	Chloroplast
*Protein metabolism*								
24	O81154	Cysteine synthase	34.32	19.3846154	*Solanum tuberosum*	−7.2	−7.2	Cytoplasm
25	Q9LST6	Proteasome subunit beta type-2	23.463	19.8113207	*Oryza sativa* subsp. *japonica*	−5.7	−5.7	Nucleus, cytoplasm
26	A2YXU2	Proteasome subunit alpha type-7-A	27.279	35.3413655	*Oryza sativa* subsp. *indica*	−7.3	0.7	Nucleus, cytoplasm
27	A1E9M6	30S ribosomal protein S8	15.7	19.1176470	*Hordeum vulgare*	−7.4	−7.4	Chloroplast
28	P51427	40S ribosomal protein S5-2	22.907	25.6038647	*Arabidopsis thaliana*	−10.4	−10.4	Cell Wall, cytosol, plasma membrane, ribosomes
29	Q949H0	40S ribosomal protein S7	22.064	16.7539267	*Hordeum vulgare*	−8.3	−8.3	Ribosome
30	P0DKK8	40S ribosomal protein S10-1	20.25	22.4043715	*Oryza sativa* subsp. *japonica*	−0.9	−8.6	Cytoplasm
31	P17093	40S ribosomal protein S11	17.822	16.9811321	*Glycine max*	−6.8	−6.8	Cytosol
32	B7F845	60S ribosomal protein L10a	24.467	20.8333333	*Oryza sativa* subsp. *japonica*	−8.1	−8.1	Cytosol
33	P42794	60S ribosomal protein L11-2	20.848	17.5824175	*Arabidopsis thaliana*	−7.3	−7.3	Nucleus, cytoplasm
34	O48557	60S ribosomal protein L17	19.494	31.5789473	*Zea mays*	−6.4	−6.4	Cytosol
35	P49690	60S ribosomal protein L23	15.017	48.5714285	*Arabidopsis thaliana*	−2.1	−9.5	Cytosol, endoplasmic reticulum, extracellular region or secreted, nucleus
36	P43643	Elongation factor 1-alpha	49.251	27.0693512	*Nicotiana tabacum*	1.5	−8.3	Cytoplasm
37	Q6EN80	30S ribosomal protein S19.	10.689	19.3548387	*Oryza nivara*	−0.2	1.7	Chloroplast
38	P50300	S-adenosylmethionine synthase	43.141	16.2849872	*Pinus banksiana*	10.1	10.0	Cytoplasm
*Carbohydrate and energy metabolism*								
39	P12863	Triosephosphate isomerase	27.008	38.3399209	*Zea mays*	−1.5	−1.2	Cytoplasm
40	P92549	ATP synthase subunit alpha	55.011	26.035503	*Arabidopsis thaliana*	0.8	−9.6	Mitochondrion
41	P00056	Cytochrome c	12.005	39.6396396	*Zea mays*	1.0	−9.8	Mitochondrion
42	P42895	Enolase 2	48.132	45.0672646	*Zea mays*	−6.4	−6.4	Cytoplasm
43	Q6XZ79	Fructokinase-1	34.669	26.0061919	*Zea mays*	−2.1	0.2	Cytosol
44	P26518	Glyceraldehyde-3-phosphate dehydrogenase	36.959	20.5278592	*Magnolia liliiflora*	1.4	−8.5	Cytoplasm
45	P26517	Glyceraldehyde-3-phosphate dehydrogenase 1	36.491	44.2136499	*Hordeum vulgare*	−10.7	−10.7	Cytoplasm
46	Q09054	Glyceraldehyde-3-phosphate dehydrogenase 2	36.519	39.7626112	*Zea mays*	1.2	−8.1	Cytoplasm
47	P93805	Phosphoglucomutase, cytoplasmic 2	63.002	39.4511149	Zea mays	−0.3	−1.6	Cytoplasm
48	O24047	Malate dehydrogenase	35.475	25.6024096	*Mesembryanthemum crystallinum*	−7.7	−7.7	Cytoplasm
49	Q9ZP06	Malate dehydrogenase 1.	35.782	18.4750733	*Arabidopsis thaliana*	−8.4	−8.4	Mitochondrion
50	P80269	NADH dehydrogenase (ubiquinone) iron–sulfur protein 8.	26.361	23.1441048	*Solanum tuberosum*	−8.8	−8.8	Mitochondrion
51	P42066	Phosphoenolpyruvate carboxykinase (ATP)	74.35	23.2835820	*Cucumis sativus*	0.8	−2.0	Cytoplasm
52	Q9XFA2	Phosphoenolpyruvate carboxykinase (ATP) 2	68.59	22.5239617	*Urochloa panicoides*	−0.7	−9.4	Cytoplasm
53	Q6YZX6	Putative aconitate hydratase	98.021	15.0334075	*Oryza sativa* subsp. *japonica*	1.5	−6.4	Cytoplasm
54	Q6ZFT5	Ribose-phosphate pyrophosphokinase 4	36.126	17.8461538	*Oryza sativa* subsp. *japonica*	1.1	−6.6	Cytosol, plasma membrane, cytoplasm,
55	Q42971	Enolase	47.942	39.6860986	*Oryza sativa* subsp. *japonica*	1.5	0	Cytoplasm
56	Q6AVT2	Glucose-1-phosphate adenylyltransferase large subunit 1	55.392	17.4168297	*Oryza sativa* subsp. *japonica*	1.1	−8.3	Chloroplast, amyloplast
57	Q6Z1G7	Pyruvate dehydrogenase E1 component subunit beta-1	39.919	15.2406417	*Oryza sativa* subsp. *japonic*	8.9	0	Mitochondrion
*Antioxidant defense and stress response*								
58	P38559	Glutamine synthetase root isozyme 1	39.226	24.6498599	*Zea mays*	−6.7	−6.7	Cytoplasm
59	O23877	Ferredoxin-NADP reductase, embryo isozyme	41.788	16.9312169	*Oryza sativa* subsp. *japonica*	0.3	−7.7	Chloroplast
60	Q6QPJ6	Peroxiredoxin Q	23.402	22.5352112	*Populus jackii*	−7.4	−7.4	Chloroplast
61	Q69SV0	Probable L-ascorbate peroxidase 8	51.156	19.2468619	*Oryza sativa* subsp. *japonica*	−0.3	−8.3	Chloroplast
62	Q02028	Stromal 70 kDa heat-shock-related protein.	75.469	24.5042493	*Pisum sativum*	1.5	−9.0	Chloroplast
63	Q6ZFU6	Thioredoxin reductase NTRB	34.655	17.2205438	*Oryza sativa* subsp. *japonica*	0.6	−8.0	Cytoplasm
64	Q9BAE0	ATP-dependent zinc metalloprotease FTSH	75.633	19.8300283	*Medicago sativa*	7.8	8.2	Chloroplast
65	P14655	Glutamine synthetase	46.613	18.6915888	*Oryza sativa* subsp. *japonica*	7.8	0	Chloroplast
66	Q01899	Heat shock 70 kDa protein	72.493	15.7037037	Phaseolus vulgaris	1.9	−9.6	Mitochondrion
67	P27322	Heat shock cognate 70 kDa protein 2	70.663	30.7453416	*Solanum lycopersicum*	9.5	8.4	Cytoplasm
68	Q08275	17.0 kDa class II heat shock protein	17.036	18.1818181	*Zea mays*	1.8	2.3	Cytoplasm
69	P27880	18.2 kDa class I heat shock protein	18.154	19.6202531	*Medicago sativa*	6.6	0	Cytoplasm
70	A5H8G4	Peroxidase 1	38.33	17.7111717	*Zea mays*	2.2	2.4	Vacuoles
71	Q9SMB1	Spermidine synthase 1	35.124	15.7894737	*Oryza sativa* subsp. *japonica*	1.8	1.3	
*Proteins involved in RNA and DNA Processing*								
72	O48556	Soluble inorganic pyrophosphatase	24.354	30.3738318	*Zea mays*	−7.7	0.3	Cytoplasm
73	Q99070	Glycine-rich RNA-binding protein 2	16.35	23.8095238	*Sorghum bicolor*	1.4	2.2	
*Proteins involved in Growth, Development and Cellular Organization*								
74	P20904	Actin	41.732	28.3819629	*Volvox carteri*	1.4	3.0	Cytoskeleton
75	P02582	Actin-1	41.591	20.2666667	*Zea mays*	8.4	9.2	Cytoskeleton
76	Q9AY76	Actin-depolymerizing factor 2	16.783	22.7586206	*Oryza sativa* subsp. *japonica*	4.3	0	Cytoskeleton, Cytoplasm
77	Q41764	Actin-depolymerizing factor 3	15.89	18.7050359	*Zea mays*	0	8.3	Cytoplasm
78	P53492	Actin-7	41.709	44.0318302	*Arabidopsis thaliana*	1.5	7.6	Cytoskeleton
79	Q41738	Thiamine thiazole synthase 1	37.081	19.7740113	*Zea mays*	−0.5	1.7	Chloroplast
*Signal Transduction*								
80	Q9SP07	14-3-3-Like protein	29.235	39.7683398	*Lilium longiflorum*	0.1	2.1	
81	Q7XTE8	14-3-3-Like protein GF14-B	29.845	64.5038167	*Oryza sativa* subsp. *japonica*	1.0	2.2	Nucleus, cytoplasm
82	Q2R2W2	14-3-3-Like protein GF14-D	29.244	39.6226415	*Oryza sativa* subsp. *japonica*	2.7	2.4	
83	P49104	Ras-related protein Rab-2-B	23.046	35.7142857	*Zea mays*	6.4	7.3	Golgi apparatus, endoplasmic reticulum
84	P28186	Ras-related protein RABE1c	23.82	22.2222222	*Arabidopsis thaliana*	−7.2	1.6	Plasma membrane, Golgi apparatus
85	O49513	Ras-related protein RABA1e	24.315	23.0414747	*Arabidopsis thaliana*	8.6	7.6	Plasma membrane
86	O24461	Ras-related protein Rab7	23.212	19.8067632	*Prunus armeniaca*	0	6.3	Plasma membrane
*Membrane transport and cell wall metabolism*								
87	P27080	ADP, ATP carrier protein	33.506	19.4805194	*Chlamydomonas reinhardtii*	0	6.0	
88	P27081	ADP/ATP carrier protein, mitochondrial (fragment)	41.802	17.8756477	*Solanum tuberosum*	0	7.4	Mitochondrion
89	P29036	Ferritin-1	28.007	18.1102362	*Zea mays*	1.7	0.4	Chloroplast, plastid
90	Q05737	GTP-binding protein YPTM2	22.461	46.7980296	*Zea mays*	6.2	7.5	Plasma membrane

^1^ Accession: accession number in the NCBI database. ^2^ MW (kDa): molecular weight of the protein. ^3^ Coverage: percentage of the protein sequence covered by the peptides. ^4^ Fold Change: a measure that describes the degree of protein change between treatments (TC50 and TC30) and the control. Only proteins with more/less than a 1.5-fold change in the abundance level or a significant difference (*p* < 0.05) were chosen. Shaded proteins showed the greatest changes in abundance (at least one treatment): down (grey) and up (green) ^5^ Subcellular Localization: the localization of the protein in a cell.

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
