# Peer review of "Proteomic Changes in Paspalum fasciculatum Leaves Exposed to Cd Stress"

_plants, 2022, doi:10.3390/plants11192455_

Round 1
Reviewer 1 Report
The manuscript concerns important topic of plant response to Cd on the proteomic level and would add new interesting data to the current knowledge. The manuscript is in general well written and enriched by informative figures.
However, I have one major concern on the design of the study – as far as I understood the Cd treated as well as control plants were grown on soil from mining site which could be already contaminated with metals. If is so, I would suggest to carry out the metal content analysis of the soil and in the case of significant level of contaminant include in the study additional control grown on optimal soil.
Other suggestions:
- The table 2 would fit better in the Results section. I would also suggest to extend the description of this Table and highlight for example with colour the proteins showing the highest changes in the abundance.
- I would suggest to delete the 3.1 subsection in the Discussion and divide the discussion solely on the subsections focused on particular groups of proteins.
- Figure 2 is very attractive but I would suggest to increase text font size
Line 71-72: I would suggest to rewrite the sentence as it suggests that antioxidant response of Paspalum fasciculatum is something unique, while it is rather usual for plants.
Some stylistic and spelling mistakes occur and therefore the text should be read carefully to correct all the linguistic issues, e.g.:
Line 59: tisular, should be tissular
Lines 64-66: All plant species are named in Latin except for the wheat. I would suggest to unify it, best of all through mentioning all names in English and the Latin names in brackets.
Lines 86-89: some stylistic issues
Line 127: I am not sure if “changes experience” is the best term in the case of proteome
Author Response
Comments and Suggestions for Authors
The manuscript concerns important topic of plant response to Cd on the proteomic level and would add new interesting data to the current knowledge. The manuscript is in general well written and enriched by informative figures.
However, I have one major concern on the design of the study – as far as I understood the Cd treated as well as control plants were grown on soil from mining site which could be already contaminated with metals. If is so, I would suggest to carry out the metal content analysis of the soil and in the case of significant level of contaminant include in the study additional control grown on optimal soil.
Ans: Metal content analysis of the soil it has been describe in reference 26. To highlight this information was added a sentence in the results and materials and methods sections, thus:
Line 82: The physical and chemical characteristics of the soil samples and the concentrations of Cd in the tissues and the growth behaviour of P. fasciculatum were established in a previous work [26].
Line 355: The preparation, physical and chemical characteristics of the soil samples have been reported in Salas Moreno et al. [26]. Besides, in table S2 are the concentrations of metals present in the initial mining soil
Other suggestions:
- The table 2 would fit better in the Results section. I would also suggest to extend the description of this Table and highlight for example with colour the proteins showing the highest changes in the abundance.
Ans: Table was moved to results section and those with highest changes were shaded in grey (down) and green (up). A sentence was added to table foot: Shaded proteins showed the greatest changes in abundance in at least a treatment: down (grey) and up (green)
- I would suggest to delete the 3.1 subsection in the Discussion and divide the discussion solely on the subsections focused on particular groups of proteins.
Ans: Recommendation was accepted and the subsections were renumbered
- Figure 2 is very attractive but I would suggest to increase text font size
Ans: Changes were realized
- Line 71-72: I would suggest to rewrite the sentence as it suggests that antioxidant response of Paspalum fasciculatum is something unique, while it is rather usual for plants.
Ans: Sentence was changed. See line 73: Similar to other plants, it also exhibits an antioxidant response which limit the oxidative damage promote by these metals on its proteome
Some stylistic and spelling mistakes occur and therefore the text should be read carefully to correct all the linguistic issues, e.g.:
Line 59: tisular, should be tissular
Ans: It was corrected
Lines 64-66: All plant species are named in Latin except for the wheat. I would suggest to unify it, best of all through mentioning all names in English and the Latin names in brackets.
Ans: The sencence was edite. See lines 66 - 68 :Java fern (Microsorum pteropus) [15], grass(Brachypodium distachyon) [3], (Arabidopsis thaliana)[18], rice (Oriza sativa) [19, 20], Populus yunnanensis [21], mustard plant (Brasica juncea) [22], canola (Brasica napus) [23] and wheat [24]. Line 65-67
Lines 86-89: some stylistic issues
Ans: The phrase was rewritten. See line 88
Line 127: I am not sure if “changes experience” is the best term in the case of proteome
Ans: The sencence was edited. See lines 129-132: The changes in the proteome of plants with phytoremediation or tolerant characteristics to Cd-stress have been the subject of many investigations because they contribute to elucidating of the resistant molecular mechanisms selected in them
Reviewer 2 Report
Although this review contains many information most of them are known. There are no critical aspects. How proteomics modifies plant physiology and metal uptake is not appropriately addressed. Also, most of the recent references on this area are overlooked.
The writing quality is very poor and there are many formatting errors.
This paper must be edited by an English professional Editor before any further submission.
Author Response
Comments and Suggestions for Authors
Although this review contains many information most of them are known. There are no critical aspects. How proteomics modifies plant physiology and metal uptake is not appropriately addressed. Also, most of the recent references on this area are overlooked.
Ans: We are not sure if the reviewer revised well the manuscript because talks about a review and not a research paper. Also the comments are very general and vague, making it impossible to deduce what to change.
The writing quality is very poor and there are many formatting errors.
This paper must be edited by an English professional Editor before any further submission.
Ans: English edition was performed.
Reviewer 3 Report
Suggested changes:
Line 21 : 30mgkg-1 -> 30 mg kg-1 or 30 mg.kg-1
Measurand an unit must be separated by a space. Different units must be separated by a space or dot. The same observation for all units.
Do not separate measurand from its units.
Line 36: Cadmium is also emitted by vehicles in circulation. In internal-combustion vehicles, it is emitted by the exhaust gases and by friction elements such as brakes and tyres. In electric vehicles only the latter type of emissions occurs.
Line 65: …Brasica 65 juncea [22] and Brasica napus [23], wheat [24]. -> …Brasica 65 juncea [22], Brasica napus [23] and wheat [24].
Figure 1: Remove border in figure. Remove border and shadow in legend values. For a more scientific presentation, graph must be planar, without 3D effect.
Figure 2: Figure legend is too long. Leave a short caption and pass the rest of the caption text to the body of the article.
References:
Some referenced articles have not indicated the doi. Please add.
Author Response
Comments and Suggestions for Authors
Suggested changes:
- Line 21 : 30mgkg-1 -> 30 mg kg-1 or 30 mg.kg-1
Ans: It was corrected
- Measurand an unit must be separated by a space. Different units must be separated by a space or dot. The same observation for all units.
Ans: It was corrected
- Do not separate measurand from its units.
Ans: It was corrected
- Line 36: Cadmium is also emitted by vehicles in circulation. In internal-combustion vehicles, it is emitted by the exhaust gases and by friction elements such as brakes and tyres. In electric vehicles only the latter type of emissions occurs.
Ans: It was corrected. See lines: 36-39, Cadmium (Cd) is a non-essential element, toxic to living organisms, it is fatal at dos-es of 1500 to 8900 mg, it is widely distributed in various environmental matrices, it is emitted by internal combustion vehicles and friction elements such as brakes and tyres.
Lines 153-154: Talebzadeh, F.; Valeo, C.; Gupta, R. Cadmium Water Pollution Associated with Motor Vehicle Brake Parts. IOP Conf. Series: Earth and Environmental Science 2021,691, 012001 doi:10.1088/1755-1315/691/1/012001
- Line 65: …Brasica 65 juncea [22] and Brasica napus [23], wheat [24]. -> …Brasica 65 juncea [22], Brasica napus [23] and wheat [24].
Ans: The sencence was edited, thus :Java fern (Microsorum pteropus) [15], grass(Brachypodium distachyon) [3], (Arabidopsis thaliana)[18], rice (Oriza sativa) [19, 20], Populus yunnanensis [21], mustard plant (Brasica juncea) [22], canola (Brasica napus) [23] and wheat [24]. Line 65-67
- Figure 1: Remove border in figure. Remove border and shadow in legend values. For a more scientific presentation, graph must be planar, without 3D effect.
Ans: Picture was modified according recomendattion. See line :126, figure 1
- Figure 2: Figure legend is too long. Leave a short caption and pass the rest of the caption text to the body of the article.
Response: The figure contains a lot of name data and is therefore extensive. We wouldn't know how to make it shorter
References: Some referenced articles have not indicated
Round 2
Reviewer 1 Report
The manuscript still needs editing as there are many grammatical, stylistic and interpunction errors.
I would suggest to move the Table S2 to the main section and describe also the methodology in methods.
As far as I unerstood the part 2.1 does not present the results of present study but of earlier findings and thus should be moved to the introduction section. I would suggest to extend this part to highlight relation to the previous experiments.
As far as I can see the lines 359-360 describe methodology of results which are not presented in the manuscript. Similarly in 4.6 biomass, Pb concentration, are mentioned but not reflected in the graphs or tables in the results.
For the consistency, the names of the plants should be mentioned either in English or in Latin or best of all with both names everywhere in the text.
Abbreviation ARN should be explained in Fig. 1 and Fig. 2.
In line 143 it should be explained what are TC30 and TC50, it would facilitated following of the discussion.
Table 1 shows proteins with changes accumulation in Cd but in description under the table the Pb is mentioned.
Author Response
Thank you very much by your revision:
Comments and suggestions for authors
- The manuscript still needs to be edited as there are many grammatical, stylistic and interpoint errors.
Ans: The manuscript was submitted for review and editing in English. See certificate attached
- I would suggest moving Table S2 to the main section and also describe the methodology in methods.
- From what I understood, part 2.1 does not present the results of this study but rather previous findings and therefore should be moved to the introduction section. I would suggest expanding on this part to highlight the relationship to previous experiments.
Ans: These data are copyrighted because they are already published in reference 26. We consider that we can leave it in the results section, but as an introduction to it. In this way, we show that it is the continuation of a deepening work on this plant species. For this, we move number 2.1 to the next section.
Consequently, the presentation of results is as follows:
The physical and chemical characteristics of the soil samples and the concentrations of Cd in the tissues and the growth behaviour of P. fasciculatum were established in a previous work [26]. These concentrations in the leaves of the plant were in a range of 4.2 a 55.72 mg kg-1, and were measured in periods of 30, 60, and 90 days. Whereas, the roots showed three times more levels of Cd than other organs (66.08 to 367.98 mg kg-1), which it is a clear mechanism of exclusion of heavy metals.
Also a higher growth was observed in the plants with treatments TC30 (30 mg kg-1 Cd) and TC50 (50 mg kg-1 Cd) with respect to control group (P≤0.05) into the first 60 days. Due these behavior, the 60-day-old plants were selected for the proteomic analysis and the results obtained are showed to next.
- As far as I can see, lines 359-360 describe the methodology of the results that are not presented in the manuscript. Similarly, in 4.6 biomass, the concentration of Pb is mentioned but not reflected in the graphs or tables of the results.
Ans: Sentence was eliminated
Along the 60-day experiment, plants were visually monitored and biomass and metal content in root, stem, and leaf determined .
- In section 4.6 you should only mention the statistics applied to proteomic analysis, because the other results have already been published.
Ans: The text has been corrected, in line 454-456
- For consistency, plant names should be mentioned in English or Latin or, best of all, both names throughout the text. Nevertheless, the changes in protein abundance under metal stress conditions should be determined to characterize specific adaptative responses in a phytoremediator plant. For this, different quantitative and qualitative proteomics techniques have been used alone or in combination to elucidate the effects of Cd stress in different plant species, as for example, Java fern (Microsorum pteropus) [15], grass(Brachypodium distachyon) [3], (Arabidopsis thaliana)[18], rice (Oriza sativa) [19, 20], Populus yunnanensis [21], mustard plant (Brasica juncea) [22], canola (Brasica napus) [23] and wheat (name?) [24].
Ans: We will only use the scientific Latin for plants to homogenize
- The abbreviation RNA should be explained in Fig. 1 and Fig. 2.
In figure 1 you use RNA and RNA for the same acronym. Homogenizes and defines.
Figure 2 includes the definition of the abbreviation.
Ans: Abbreviations have been corrected in the text and figures.
- In line 143 it should be explained what TC30 and TC50 are, it would facilitate the follow-up of the discussion.
Ans: The modification was made as requested:
To explain these tolerance mechanisms, we focused on the behavior of the 90 proteins that showed statistically significant abundance changes in P. fascicultaum leaves in the TC30 (30 mg kg-1 Cd) and TC50 (50 mg kg-1 Cd)…
- Table 1 shows proteins with accumulation of changes in Cd but in the description below the table Pb is mentioned.
Ans: Sentence was corrected: 4Fold Change: It is a measure that describes the degree of protein change between treatments (TC50 and TC30) and control.

Reviewer 2 Report
As I mentioned earlier this article is poorly written.
Both results and discussion are wordy and results are unclear.
There are many formatting issues.
Abbreviations and symbols should appear at the beginning of the sentence. e.g. Cd should be Cadmium.
Many of the references are old. There are many recent references on Cd but these are overlooked.
Author Response
Thank you by your revision:
- As I mentioned earlier this article is poorly written.
- Both results and discussion are wordy and results are unclear.
- There are many formatting issues.
Asn: The manuscript was submitted for review and editing in English. See certicate attached.
- Abbreviations and symbols should appear at the beginning of the sentence. e.g. Cd should be Cadmium.
Ans: following instructions to authors, abbreviations and symbols are only placed once at the beginning of the text
- Many of the references are old. There are many recent references on Cd but these are overlooked.
Ans: References have been updated, references: 1, 6, 10, 11,12, 13, 14, 16, 18, 27, 33, 34, 44, 45, 46, 47, 53, 54, 55, 57, 58, 59, 60

Round 3
Reviewer 2 Report
As I mentioned earlier and now i am quite sure that the authors are not really careful about writing quality and do not have any interest in such a revision.
For example, the results section starred with a discussion: The physical and chemical characteristics of the soil samples, the concentration of Cd in the tissues, and the growth behavior of P. fasciculatum were established in previous work [26].
There should be only results without any discussion or references.
The entire manuscript is now warrant professional language editing, formatting and further submission for review. Otherwise, it may be rejected.